# Adaptation of the Four Levels of Test Maturity Model Integration with Agile and Risk-Based Test Techniques

**Ahmet Unudulmaz [1,2,\*], Mustafa Özgür Cingiz [3] and Oya Kalıpsız [1]**

1    Computer Engineering Department, Yildiz Technical University, Istanbul 34200, Turkey; kalipsiz@yildiz.edu.tr
2    Research and Development Department, Siemens, Istanbul 34870, Turkey
3    Computer Engineering Department, Bursa Technical University, Bursa 16310, Turkey; mustafa.cingiz@btu.edu.tr
\*    Correspondence: ahmetunudulmaz@gmail.com

**Abstract:** Many projects that progress with failure, processes managed erroneously, failure to deliver products and projects on time, excessive increases taking place in costs, and an inability to analyze customer requests correctly pave the way for the use of agile processes in software development methods and cause the importance of test processes to increase day by day. In particular, the inability to properly handle testing processes and risks with time and cost pressures, the differentiation of software development methods between projects, the failure to integrate risk management, and risk analysis studies, conducted within a company/institution, with software development methods also complicates this situation. It is recommended to use agile process methods and test maturity model integration (TMMI), with risk-based testing techniques and user scenario testing techniques, to eliminate such problems. In this study, agile process transformation of a company, operating in factory automation systems in the field of industry, was followed for two and a half years. This study has been prepared to close the gap in the literature on the integration of TMMI level 2, TMMI level 3, and TMMI level 4 with SAFE methodology and agile processes. Our research has been conducted upon the use of all TMMI level sub-steps with both agile process practices and some test practices (risk-based testing techniques, user scenario testing techniques). TMMI coverage percentages have been determined as 92.85% based on TMMI level 2, 92.9% based on TMMI level 3, and 100% based on TMMI level 4. In addition, agile process adaptation metrics and their measurements between project versions will be shown, and their contribution to quality will be mentioned.

**Keywords:** TMMI; agile; SAFE; testing techniques; risk-based tests; test metrics; quality; quality management; TMMI level 2; TMMI level 3; TMMI level 4

## 1. Introduction

As it also takes place in the agile process manifesto, customer collaboration has stood out on the foundation of agile methodology, rather than individuals and interactions, processes and tools, employee working contract negotiations, and comprehensive documentation [1,2]. Agile process testing is a software development model in which requirements can change dynamically, and customers can be involved in software development and testing processes. The testing stage needs to be handled from the start of the project to the end and progress in parallel with software development in the agile process. In the context of quality, test load and test activities are key areas for projects. Today, in the projects where agile process concepts are used, testing processes are generally neglected, tests and results are not followed up, and no process development model or test concept is used or used less in the view of testing. It is aimed to adapt the Test Maturity Model (TMMI) to cover this gap in agile processes. It is known that, when TMMI is adapted to projects, it has a positive impact on the project and product quality, improves test processes, and reduces test load and effort [3,4]. Along with the said benefits, documentation and

limitation burden comes with TMMI as well. When used with agile processes, it is foreseen that this documentation burden will be eliminated. The first three levels—TMMI level 2, TMMI level 3, and TMMI level 4—will be addressed for the integration of the test maturity model (TMMI) with agile processes. As the agile process model, it is considered to use the SAFE model that takes place under the large-scale agile process models. In addition to these studies, it is aimed to use the risk-based testing approach. Thanks to it, it is predicted to facilitate the integration of agile processes and TMMI, as well as increase product quality.

This study differs in several points when the studies in the literature are examined. In literature, it is observed that addressing TMMI level 2 with agile processes is discussed. There is not any study on the integration of large-scale scrum and TMMI. In addition to TMMI level 2, there is SAFE integration of TMMI level 3, and TMMI level 4 is also integrated in our study. Company data were obtained after the agile transformation process and were analyzed in our study. In addition, the risk-based testing technique has been harmonized with agile processes in order to increase the coverage rate of integration. Measurements were applied by using various metrics among the project release versions of the company. In each project release version, a different TMMI level is adapted to the project, and the differences between both project versions and TMMI levels were analyzed.

In our contributions, we have:

- To ensure the addressing of the first four levels of TMMI with SAFE and to prove its applicability.
- To increase coverage rates by using risk-based testing techniques during the mapping phase and integrate these techniques into scrum rituals.
- To aim to provide a detailed measurement based on the test scenario, defect quality, and test coverage by using several metrics between agile processes and the integration of TMMI versions.
- To prove the applicability of the model in large-scale companies by comparing the measurements of our study with both TMMI versions and other studies in the literature.

The rest of the study is composed as follows. Section 2 presents a literature review on large-scale scrum processes, integration studies with TMMI and agile processes, and risk-based testing approaches, which will also be used within the scope of the study. In Section 3, the implementation stages will be mentioned. Accordingly, after mentioning the research approach and assumptions, the implemented project and team structure will be specified, and it will continue with case analysis. Under the case analysis, addressing of the sub-areas of TMMI level 2, TMMI level 3, and TMMI level 4, with agile processes and risk-based testing approaches, will be mentioned. In Section 4, addressing agile processes with TMMI levels and project measurement results with metrics will be handled, and information regarding the impact of these studies on the quality will be shared. Section 5 includes a detailed comparison of the data obtained as a result of the measurements. In addition, the accuracy of the application will be mentioned by comparing it with the state-of-the-art application. Section 6 concludes the paper and highlights future work in this area of research.

## 2. Related Studies

Related studies will be examined under three categories that are intended to integrate into our study. The first of these will be the studies on TMMI. TMMI and different test maturity models will be examined in this part. Comparison of these models with TMMI and their integration with agile processes will also be included. The second part consists of studies on large scale scrum. In this part, the team structure and applicability of large-scale scrum, applied in different companies, will be mentioned. In the third part, risk-based approaches and studies in this field will be presented.

### 2.1. TMMi

Researchers conducted a literature review on software testing process improvement models at Malardalen University. Then, 18 different test improvement models were scru-

tinized. Subsequent to that screening, TMM, TMMI, TPI Next, Test Spice, and TPI Next models were compared in detail. In the second stage of the study, the sub-areas of TPI Next and TMMI models were also addressed, and the differences between these two models were the focus. While TMMI offers a progressive model, TPI Next offers a continuous model [5]. In the study conducted at Hacettepe University, a literature screening was conducted on test process improvement models in parallel with the study conducted at Malardalen University. Test maturity model (TMA) and test process improvement (TPI)-based models were compared across the screening. A total of 58 different process improvement models were used [6]. According to the documents published by the TMMI foundation, a study on the joint usability of agile processes and TMMI has been published. The sub-areas that are compatible with agile processes, on the basis of TMMI levels, were specified. Accordingly, they expressed that, on the level two basis, the steps of test policy and test strategy, test planning, test monitoring, test design and run, and test environment are highly adaptable to agile processes [7,8]. In the study conducted at Helwan University, they conducted research on agile process practices, addressing the second level of TMMI and the performance, as well as quality factors in the project. According to that study, all test sub-areas matched with agile process practices. In the test configuration section, they found a 90% mismatching, or semi-applicable, area [9].

### 2.2. Large Scale Scrum

Along with the studies on TMMI, particular studies have also been compiled, through application examples, on large-scale scrums and companies. Accordingly, in a study conducted at Spotify, the concept of application community was introduced with agile processes in a company. An application community (AC) usually consists of a group of people with similar skills and interests who share knowledge, make joint decisions, solve problems jointly, and develop an application. Application groups form a value for individuals, teams, projects, and the organization as a whole, while influencing the knowledge culture and developing an application. Information exchange in ACs is achieved by means of a variety of planned and unplanned forms of social interaction, such as meetings and conferences, brown lunches, newsletters, teleconferences, email lists, discussion forums, and simultaneous chats. This study specifies findings from scrutinizing member engagement in large-scale distributed application communities, called guilds, on Spotify. Accordingly, studies were conducted in eight different Spotify guilds. It shows that maintaining successful large-scale dispersed guilds and active participation are a real challenge. It was observed that only 20% of the members participate in guild activities regularly, the majority only subscribe to the latest news, and the guild structure and topics were arranged accordingly. Organizational size and distribution became the source of many obstacles before participation. Possession of too many members and, particularly, time-based distance means that scheduling joint meeting times is problematic. In the agile process organization, such situations must be taken into account, and guilds must be made from appropriate individuals and project groups [10,11].

### 2.3. Risk Based Approaches

In addition to TMMI and large-scale agile process studies, risk-based approaches were examined. Accordingly, risk assessment in software projects ensures the determination of risks and risk factors that may take place in the project. Risk factors regarding issues such as budget, time, and labor lead to uncertainty in the requirements analysis stage, which is the beginning of the project. Therefore, software project risk management has to be conducted. The right model and method must be determined while risk management is being performed. Even if the ratio of determination of the risks of the project in software projects is 90%, according to IEEE, it was observed, in studies, that the ratio of identification of risks is between 50–70%. The return of risk management as an investment is at a value between 700% and 2000%. When the time and labor spent are calculated, it is mandatory that the risks and their consequences are identified beforehand, and precautions must be

taken according to the importance and probability of the risks [12,13]. In their study in question, Kitchenham and Linkman addressed the issues of estimation, uncertainty, and risk. It was highlighted that estimation models must be created by taking such factors into account, and it was stated that assumptions must be made for unidentified risk factors. It was mentioned that assessment models can be drawn by taking the impact values of erroneous assumptions [14]. Kwan and Leung addressed the risk management methodology and project risk dependencies in their study. A risk dependency graph was created by means of using the conservative method, the optimistic method, and the weighting methods. After the risk graphs were formed, risk management paradigms were addressed in view of risk dependence. Accordingly, criticality levels were determined for three different situations. The first of them is the level of internal severity. It is directly used in risk management paradigms. The second is the severity level used for the occurrence of the other risk. As the third, the response severity level is used. Different action plans are made according to the levels [15]. Ray and Mohapatra have published a study on risk analysis. They worked on the case-based risk prevention methodology for the analysis and design steps in the software development lifecycle. They reduced the complexity between components and scrutinized each component with graph methods by developing a risk estimation method. In addition, they analyzed the failure scenarios at system levels by performing a criticality analysis. Accordingly, they brought together the defect leveling, in four levels, as catastrophic, major, marginal, and minor [16]. In addition, a few more articles in this field have been reviewed to conduct risk-based testing and incorporate test automation into the process. Details of the use of different test tools, for functional and non-functional tests, are specified [17,18].

## 3. Materials and Methods

Agile methodology refers to practice-based methods for modeling and documenting software systems effectively and efficiently. It offers easy and flexible solutions to execute complex properties and respond to customer requirements quickly. It provides transparency, control, and adaptation during software development to avoid complexity. Nevertheless, the scrum guide does not provide details in terms of documentation on testing and quality, as it is a "frame" for developing, presenting, and maintaining complex products [2]. Moreover, as a result of dynamic changes in product and customer requirements, complete testing becomes impossible, and tests must be prioritized in line with customer needs. From a different perspective, using TMMI in a manner that is integrated with a different software development model inflicts some problems. Along with the heavy documentation burden that comes with TMMI, TMMI adaptation in smaller organizations necessitates additional resources and knowledge [7,8].

### 3.1. Problem Definition

Companies continuously try to improve their testing and quality processes by applying new models, methodologies, and techniques. TMMI can be adopted as a model in that improvement process. However, in this case, it requires a serious effort and cost, in view of time and human resources, for the implementation of TMMI. At the same time, detailed knowledge about the model is necessary as well. The fact that the corporate culture is suitable for it and that the created model is wanted to be used across the company also brings along a need for creating a flexible model. That is why the number of companies that integrate TMMI into their processes all over the world is not many. In addition to this model, making a stand-alone agile process transformation also brings along setbacks from time to time. Trying to combine TMMI with agile processes would be a good choice to avoid such disadvantages. It is necessary to make sure that the agile processes and the TMMI model can be operated in harmony with each other. If there are sub-areas or missing sub-areas that will create incompatibility, they must be addressed with different methodologies and techniques. It is necessary to analyze the integration of these concepts,

determine the coverage ratio of addressing after the work to be carried out, measure the processes with quality metrics, and demonstrate the quality.

### 3.2. Company Description

In the light of this information, it was decided to apply the SAFE model in the agile transformation process of a company, which developed a project with V-Model previously. Under the V-Model, the corresponding testing phase of the development phase is planned in parallel. There are Verification phases on one side of the 'V' and Validation phases on the other side. With this model, the operation, maintenance, organization, and execution of the system cannot be provided. In addition, since it could not be adapted to all organizations and was handled in the software development within the project, the need for a different model on a large scale emerged. In this context, a company that is in the agile process transformation process and working on plant automation and embedded systems was selected, and a three-year process was observed. The firm has specialized in embedded systems and factory automation systems. This study will be put into practice on a project team of 80 people. With the agile process transformation, the said team will be divided into eight sub-teams and will continue to develop the projects. This endeavor will be started with the agile process transformation process. Thanks to it, the opportunity to observe and measure the quality processes of the company with metrics before and after using TMMI and agile processes will be seized. It is expected that the agile process teams will consist of eight people and there will be 10 teams in total. The teams will consist of the product owner, scrum master, software team, and team architects. In addition to the teams, the roles of system architect, test architect, DevOps superintendent, and release train engineer will take part in the project by using the SAFE model as well. The first four levels of TMMI will be adapted, in an integrated way, with SAFE. Numerous sub-elements under TMMI and SAFE will also be addressed using risk-based testing approaches, use-case testing techniques, and DevOps approaches. It will aim to provide a common denominator about the test infrastructure and methods used in the projects, as well as create a template that can be applied irrespective of the size of the project and the number of people/teams. Thanks to it, developments in both project quality and product quality can be measured with this model, and its compatibility with agile processes can be revealed.

### 3.3. Research Approach

The aim of this research is to show how agile processes and the TMMI model can be used, together, and help enterprises and institutions that will operate agile processes enhance their project and product quality by using the TMMI model, in a manner that will cover the test and risk processes. The study to be conducted will, basically, consist of three stages. The first step is to perform matching between the testing process and SAFE. As can be seen in Table 1, processes and rituals in scrum are handled through software development stages without a separate process [8].

In the second stage, the matching of the sub-areas of TMMI level 2, TMMI level 3, and TMMI level 4 with agile processes are, theoretically, present. This method will be used to address TMMI levels as agile process practices. By addressing each sub-level of the model one by one, the extent that the sub-steps and agile process sub-steps overlap with each other will be studied. In order to be able to do the study and increase the addressing rate, a risk-based testing approach, DevOps approaches, and user scenario testing techniques will be used in addition to the agile processes. As stated in Table 2, if TMMI practice can be implemented with agile process and test process practice, it will be matched with "Yes, it is applicable (Y) level". If some areas are applied and some areas are not applied—that is, if it is not fully suitable for agile processes but may take place with some practices—it will be matched with "Semi-applicable (SA) level". If it cannot be matched with any practice, it will be matched with the "No, not applicable (NA)" level.

**Table 1.** Test Processes and SAFE Activity Integration.

| Agile Process Activities | Participants | Entry Criteria | Exit Criteria | Test Process |
|---|---|---|---|---|
| Refinement Meeting | Scrum team + software and test architect | User scenarios + PBI | Risks, Test levels, and test technical decisions | Requirements analysis |
| Planning Meetings | Scrum team + software and testing architect | User Scenarios + PBI + Risks | PBI Assignment, User scenarios | Test Planning |
| Daily Stand-Up Meetings Other sprint days | Scrum team | User Scenarios + PBI + Risks | Test Scenarios, Test Automation code, Test Environment, Test results, Defects | Test Scenario development Test Analysis and Design Test Environment Setup Test Execution |
| Review Meeting | All teams + All Architects + DevOps + Product managers | Test Automation code + Test results + Defects | Revised Product Backlog | Test Closure |

**Table 2.** Matching Criteria.

| Remark | Criterion |
|---|---|
| Practical applicable | Yes applicable (Y) |
| Practical semi applicable | Semi applicable (SA) |
| Practical not applicable | No not applicable (NA) |

In the third stage, the measurements made at all TMMI levels will be compared with each other and the project and product before and after scrum-TMMI adaptation impacts on quality will be shown. In addition to the metrics recommended for use under TMMI level 4, metrics used under scrum and metrics used in project risk management are planned to be included. Adaptation results, based on metrics and TMMI levels, will be conveyed, in detail, under the title of metric general results.

*3.4. Research Assumption*

As a result of the research, it is assumed that the use of TMMI with agile processes will have two major impacts. They are:

- With the process transformation, the increase in quality and efficiency upon the re-start of previously neglected test processes and, in parallel to them, the performance of more efficient tests with less labor and an increase in the in-team harmony.
- Upon inclusion of TMMI in agile processes, getting rid of the heavy documentation burden arising from TMMI, ensuring that the test processes become more formal and specific.

*3.5. Study Phase*

The theoretical matching of the sub-areas of TMMI level 2, TMMI level 3, and TMMI level 4 with agile processes will be scrutinized under this title. While making these matches, risk-based testing techniques, user scenario testing techniques, and DevOps approaches will be used, additionally, and integrated into existing models to increase both quality and the matching percentage. This method will be used to address TMMI model levels as agile process practices. By addressing each sub-level of the model one by one, the extent that the sub-steps and agile process sub-steps overlap with each other will be studied.

3.5.1. TMMI Level 2

TMMI level 2 sub-areas have been addressed within the scope of the study. Accordingly, test policy and test strategy, test planning, test monitoring and control, test design,

and test environment sub-areas will be addressed, and the compatibility of these fields with agile processes is examined.

- When the test policy and test strategy sub-areas are taken into account, many sub-areas have been addressed with agile processes, thanks to the sprint planning meetings, under the agile processes, and the release planning meetings, held at the onset of the project, in order to determine the test targets. In parallel with this addressing, the determination of the test policy and test strategy was also addressed with the said meetings and the refinement meetings to be held in each sprint. In addition, owing to the risk-based testing approach used, the risk assessment area, which is a sub-area of the testing strategy, was also addressed. Thanks to these methods used, all sub-areas under the test policy and strategy are covered through agile processes.

- When the test planning sub-area is taken into account, the importance of the risk-based testing approach stands out. As shown in Table 1, during the determination and analysis of product risks, risks are discussed within the scope of the review, planning, and refinement meetings held for each sprint, and they are addressed, together, with the related works in Figure 1. As can be seen in Figure 1, risk identification and risk analysis steps begin with refinement and planning meetings. The scrum team, product owner, scrum master, team architect, and test architect participate in these meetings. In these meetings, while the requirements are analyzed with the team, the relevant risks are defined. Probability and likelihood values of risks are determined. After the multiplication of these values, the risk priority number emerges. Risks are prioritized according to the risk priority number. At the planning meetings, these risks are addressed with test plans, and they are aimed to be resolved within the sprint. Risks are mitigated by testing design and test execution during the remaining days of the sprint. In sprint review meetings, the relevant risks are reviewed again, and the relevant risks are closed in accordance with the definition of done (DoD).

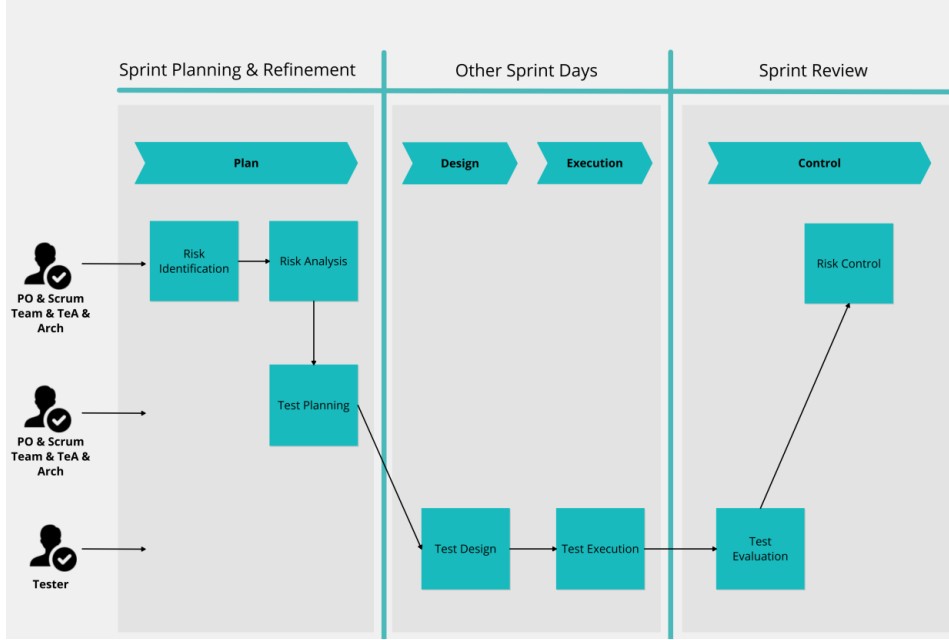

**Figure 1.** The risk-based testing approach in agile processes.

In addition to the risk-based testing approach, entry and exit criteria, which are the most important sub-areas of the testing strategy, must also be identified. These criteria must be determined by taking the ideas of the teams, and these criteria must be put into practice according to the Definition of Ready (DoR) and Definition of Done (DoD) definitions under agile processes. It must be constantly checked whether the team complies with these criteria when starting and finishing the work. Along with the entry/exit criteria, it is also necessary

to make time, resource, and person planning and ensure test plan commitments. Planning and review meetings, held under agile processes, directly overlap with the said items and are addressed. If sub-areas that cannot be addressed with agile processes and a risk-based testing approach are taken into account, then suspension and restart criteria become in question. Problem areas are addressed in daily stand-up meetings held under agile processes, but there are no suspension or restart criteria under agile processes. Hence, this sub-area could not be addressed. In addition, the sub-area of involvement of stakeholders in planning activities is marked as semi applicable when all projects are considered.

- When taking the test monitoring and control sub-area into account, it is aimed to be able to monitor the test progress and product quality. Sprint boards are followed up through daily stand-up meetings and sprint review meetings to monitor test planning, test commitments, and test progress. Project and product risks are also monitored in such a way, and risk mitigation steps are constantly monitored within the sprint by using test automation and related tools. In view of the entry/exit criteria, if work is to be started on a team basis or if that work is to be closed, full compliance with the said criteria is expected, and it is followed up. When it comes to the sub-areas that cannot be addressed with agile processes, the follow-up of the suspension and continuation criteria takes place in parallel with the test planning activity. In addition, as there is no separate meeting within the scope of agile processes for the examination of product quality and its follow-up on the basis of milestones, these sub-areas could not be addressed as well. In addition, the sub-area of monitoring the participation of stakeholders is marked as semi applicable when all projects are taken into account.

- When the test design and execution sub-area are taken into account, a risk-based testing approach was used in order to prioritize test cases and test procedures, as mentioned in the planning area. In addition to the software development process, the user scenario testing technique was used to reveal the test procedures as well. This way, the costs are reduced by beginning the test at the very start of the process, and it is ensured that the work is understood by all stakeholders. Boards are used in agile processes and defect monitoring systems, while test automation tools are also used in the test run and reporting of findings. The sub-areas of obtaining specific test data and obtaining the necessary test data could not be completely addressed. These sub-areas are marked as semi applicable as test data is not needed for every project.

- The test environment sub-area contains the respective test data to ensure that the tests can be run properly and repeatedly. Here, testing techniques and agile processes are not enough alone. In addition to them, DevOps approaches were also used. Thanks to them, many sub-areas have been addressed through this approach. The non-addressed area appears to be the analysis of the test environment requirements. In other software development methods, the test architect or test manager is responsible for the analysis of the requirements but in agile process projects, this sub-area is marked as non-addressable in case a person with this role and responsibility is not present in the team structure. The test data requirement is also marked as semi applicable, which is in line with the test design and run sub-area.

3.5.2. TMMI Level 3

TMMI level 3 sub-areas are addressed within the scope of this section. Accordingly, test organization, test training program, test life cycle and integration, non-functional tests, and stakeholder assessment sub-areas will be addressed and the compatibility of those fields with agile processes is examined.

- When the test organization sub-area is taken into account, it appears to be parallel to the sub-areas of test policy and test strategy identified in TMMI level 2. The team structure, the number of people, and the type of organization are identified. Test roles and test responsibilities, which are determined separately in the traditional structure, are not determined separately in the agile process but in a manner that the entire team will be responsible for. Accordingly, the definition and integration of the test

organization, receipt of commitments, and making of test definitions are identified by the meetings and teams within the agile processes. Rather than roles such as test manager and test team leader in the hierarchical structure, tests and quality were left to the team responsibility, and roles at the horizontal level were identified. All sub-areas of the testing organization are addressed in an accurate fashion with agile processes.

- When the test training program sub-area is taken into account, scrum masters, together with the teams, play an active role in determining the pieces of training and preparing the training plan. According to the scrum guide, the scrum master must help the team produce high-value products. While determining the test training requests, the team must determine the requests with the scrum master. Areas other than the sub-areas of keeping test training records and evaluating test training effectiveness are addressed with agile process practices. These sub-areas have no equivalent in agile processes.

- When the test life cycle and integration sub-area is taken into account, the team structure and size were also considered, and addressing was made with the SAFE methodology by making use of agile processes. Thanks to it, the possibility for many teams to work on the same project, with a fully integrated software development lifecycle model instead of a separate testing process, was provided. The Scaled Agile Framework® (SAFE) is an online knowledge base comprising proven and integrated policies, practices, and competencies to implement lean, agile, and continuous integration at a large scale. For building enterprise solutions, a single agile process frequently requires more scope and skills than a team can provide. It typically requires highly diversified expertise skills that cannot be found in a single agile process team. Thus, multiple agile process teams must collaborate. SAFE's Agile Release Train (ART) is referred to as a whole long-lived agile team that, together with other stakeholders, gradually develops, delivers, and, where appropriate, runs one or more solutions [19].

- The test process, test life cycle, and adaptation criteria were integrated into the entire process by using the SAFE methodology. The determination of the working area standards among the sub-areas could not be addressed with the test processes and SAFE methodology. With the use of the SAFE methodology, additional roles and responsibilities such as Release Train Engineer, Solution Train Engineer, System Architect, and Test Architect have also been included in the process.

- When we consider the non-functional testing sub-area, this sub-area has been handled and addressed in TMMI 2, as the risk-based testing approach is used. The test strategy, test planning, and test monitoring sub-areas addressed in TMMI level 2 take place in not only functional testing but also non-functional testing. The risks and preventive steps of the current tests are addressed within the sprint, and the results are observed at the end of each sprint. Thanks to the addressing and joint use of TMMI 2 with agile processes and risk-based testing approaches, this sub-area has been completely addressed even before reaching TMMI level 3.

- When taking the stakeholder assessment sub-area into account, the refinement, which is available under agile processes, is integrated into the process thanks to daily stand-up meetings and review meetings. The only area that could not be integrated here is the analysis of stakeholder assessment data. In the review meetings, the works related to the team are reviewed but the stakeholder assessments are not scrutinized separately.

### 3.5.3. TMMI Level 4

TMMI level 3 sub-areas are addressed within the scope of this section. Test measurement, product quality improvement, and advanced reviews of sub-areas were addressed, and the compatibility of those areas with agile processes was examined.

- Setting goals are addressed with the simplicity principle of agile processes directly. Targets are set at planning meetings. In addition to the metrics recommended under TMMI level 4 such as test effort, the number of test scenarios, defect counts, defect density, code coverage, requirement coverage, reliability measures, additional metrics such as defect quality, risk identification effectiveness, and risk reduction effectiveness

were used for risk monitoring. Thanks to it, all sub-areas are addressed with the SAFE method and agile processes. Details of the respective metrics and test measurement values are provided in the results and discussions section.

- When we consider the product quality development sub-area, there is no difference between the classical method and agile processes in this respect. The quality requirements are also determined in a manner that they comply with the DoR and DoD definitions and they also comply with the input-output criteria of the test strategy. Accordingly, when determining product quality requirements and making measurements throughout the product life cycle, progress is made by complying with the criteria determined above and by continuously measuring with metrics. Hence, all sub-areas are compatible with agile processes.

- When we look at the advanced reviews sub-area, it parallels the stakeholder reviews handled at TMMI level 3. Work products are addressed in planning meetings and refinement meetings, and it is determined at what level and how they will be tested with the team. Code quality and stakeholder review were also added to the respective metrics, and all sub-areas were addressed with agile processes.

## 4. Results

Within the scope of the study, integration of the steps under test maturity model integration (TMMI) level two, level three, and level four, with agile processes and risk-based testing approaches, were mentioned, and the coverage ratios of the levels were determined. As can be seen in Table 3, the number of addressable sub-areas, the number of non-addressable sub-areas, the number of semi-addressable sub-areas, together with the coverage ratios, are detailed on the basis of all levels.

**Table 3.** Addressing TMMI levels with Agile Processes.

| TMMI levels | Number of TMMI Subdomains | Practical Applicable | Practical Semi Applicable | Practical Not Applicable | Coverage Rate |
|---|---|---|---|---|---|
| TMMI level 2 | 70 | 65 | 0 | 5 | 92.85 |
| TMMI level 3 | 56 | 52 | 0 | 4 | 92.9 |
| TMMI level 4 | 21 | 21 | 0 | 0 | 100 |

As can be seen in Figure 1, in the first line, the integration of steps under test maturity model (TMMI) level two with agile processes is mentioned. A total of 70 sub-titles were examined under five titles. Among them, 65 could be addressed with agile processes, risk-based testing approach, and user scenario testing approach. Five of them could not be addressed with any scrum practice. As mentioned under TMMI level 2, they are the sub-areas of suspension and restarting criteria, monitoring of customers' meeting attendance, and quality reviews of product quality milestones, which are not included under agile processes. In addition, in view of the test environment title, agile processes, alone, are not sufficient to address all sub-items. For example, as mentioned in the test environment sub-item, software operation approaches were required to be included in that process. Due to it, the TMMI coverage percentage was determined to be 92.85%.

When it comes to the second line, it is mentioned that the steps under test maturity model (TMMI) level three are integrated with agile processes. A total of 56 sub-titles were examined under five titles. Among them, 52 could be addressed with agile processes, risk-based testing approach, and user scenario testing approach. Four of them could not be addressed with any scrum practice. They are keeping the test records under the test training program, evaluating the test training effectiveness, and creating working environment standards. The TMMI coverage percentage was determined to be 92.9%, as can be seen in Table 3.

When we examine the last line, integration of the steps under test maturity model (TMMI) level four with agile processes was mentioned. A total of 21 sub-titles were examined under three titles. Among them, 21 could be addressed with agile processes and

a risk-based testing approach. The TMMI coverage percentage was determined to be 100%, as can also be viewed in Table 3.

*General Results of the Metrics and Verification of the Study*

Within the scope of our study, the metrics recommended in the TMMI level 4 were used. In this way, we had the opportunity to compare our results with the documents published by the TMMI organization and with other studies in the literature [7,8]. In addition, risk assessment metrics recommended to be used in agile projects are also included in the scope of the study. In this way, the identification and monitoring of risks are integrated into the process [20]. These metrics were applied between project versions. Accordingly, the metrics used were listed as follows:

- Total number of runs: The time between the versions of the product is expressed by the runs used under agile processes. The period of the runs was determined as two weeks. Before TMMI and agile processes, one-year, that is a 24-run period, was addressed, and basic comparisons were made based on that data.
- Total defect numbers: It is the list of the total number of defects found throughout the project version. At the same time, the defects found on a monthly basis are also listed in the same vein.
- Total defect numbers by priority: Defects were grouped under three main titles. Accordingly, according to the order of importance of the defects, all defects are shown as a percentage, as critical, medium, and low.
- Test effort and cost: The number of people working as testers in agile process teams or doing the testing effort in the team divided by days was used.
- The total number of test scenarios: It means the total number of test scenarios performed on the basis of project versions. This result will be calculated cumulatively between project versions.
- The valid number of defects: They are the values obtained as a result of subtracting canceled defects or rejected defects from the total number of defects.
- Test scenario effectiveness: It is the result obtained by dividing the defects found with the test scenarios by the total number of valid defects.
- Defect quality: It is the ratio obtained by dividing the number of valid defects by the total number of defects.
- Risk identification effectiveness: It is a metric added for using the risk-based testing approach. It is the ratio of identified risks to total requirements.
- Risk reduction effectiveness: It is a metric added for using the risk-based testing approach. It is the ratio of closed risks to total risks.
- Code coverage: The ratio of automatized test scenarios to requirements.
- Weekly test run rate: It is the weekly/sprint-based run ratio of automatically running test scenarios.
- The number of postponed defects: The number of defects that cannot be resolved throughout the project version and are postponed to the next project version.
- The number of lines/number of files: It is a metric added to understand the code added throughout the project version. It is the number of files added and changed.
- The number of defects coming from the customer: It is the total number of defects that were not found in the project version and came from the customer.

Project versions were measured with the metrics specified in Table 4, and the improved areas were disclosed before and after TMMI. Measurements were made in a total of two years and four months and four different project versions. Before TMMI and agile process adaptation, one-year project data was taken into account, and comparisons were made over them. By making a new measurement subsequent to each TMMI level integration, a comparison with the previous and next versions was made possible.

**Table 4.** Measurement of the metrics after TMMI—Agile process adaptation.

| Metrics | TMMI & Pre-Agile Version 1 | TMMI-2 Version 2 | TMMI-3 Version 3 | TMMI-4 Version 4 |
|---|---|---|---|---|
| Total run times | 1 year (24 runs) | 6 months (12 runs) | 4 months (8 runs) | 6 months (12 runs) |
| Total number of defects | 110 | 52 | 20 | 30 |
| | 1 month = 9.16 defects | 1 month = 8.6 defects | 1 month = 5 defects | 1 month = 5 defects |
| Total number of defects by priority | 43% Critical, 50.9% Medium, 63% Low | 21.5% Critical, 67.3% Medium, 7.69% Low | 45% Critical, 50% Medium, 5% Low | 40% Critical, 57% Medium, 3% Low |
| Test effort and test cost | 10 man/day | 5 man/day | 5 man/day | 4 man/day |
| Total number of test scenarios | 787 | 179 | 208 | 299 |
| Number of valid defects | 76 | 45 | 19 | 29 |
| Test scenario effectiveness | 65 | 80% | 79% | 75% |
| Defect quality | 69 | 87% | 95% | 97% |
| Risk detection effectiveness | 0 | 80% | 60% | 86% |
| Risk reduction effectiveness | 0 | 50% | 12% | 53% |
| Code coverage | 32 | 43% | 37% | 40% |
| Weekly test execution ratio | 50 | 58% | 51% | 59% |
| Number of postponed defects | 14 | 2 | 5 | 4 |
| Number of lines/Number of files | 602 added, 187 changed | 804 added, 133 changed | 283 added, 140 changed | 245 added, 269 changed |
| Number of defects coming from the customer | 12 | 5 | 6 | 4 |

## 5. Discussion

In the discussion part, the coverage rates of TMMI and agile processes will be discussed first, and these rates will be verified by existing studies. The results of the TMMI level 2 and SAFE adaptation are similar to other studies in the literature. While the coverage rate was found to be 90% in a similar study, this rate was determined as 92.85% in our study [9]. This comparison confirmed that TMMI level 2 can be addressed with SAFE. When we examine TMMI level 3 results, a comparison was made with the document published by the TMMI Foundation as a baseline. According to the TMMI Foundation, this rate has been determined as 98.07%. In our study, it was found to be 92.9%, which is close to TMMI Foundation rate [8]. This comparison has confirmed that TMMI level 3 can be addressed with SAFE. Fields not addressed under test training will be studied separately. Similar to TMMI level 3, we cannot compare the literature results to our study for TMMI level 4, due to lack of studies. The comparison was only made with the document published by the TMMI Foundation for TMMI level 4. According to the TMMI Foundation, this rate has been determined as 100%. In our study, it was also found as 100% [8]. In this way, it has been verified that all subdomains of TMMI 4 can be addressed with SAFE. With our study, it has been ensured that the four levels of TMMI can be addressed with SAFE, and its applicability has been proven.

After defining coverage ratios, metric measurements and their results are discussed. We encountered several limitations when comparing measurement results. We cannot compare the literature results with our results at TMMI level 3 and Level 4 due to a lack of studies. Therefore, only TMMI level 2 comparison will be discussed, and TMMI level 3 and TMMI level 4 levels will be compared within themselves and the contributions to quality will be revealed. We compared our study to the study of Ahmed et al. [9]. In the mentioned study, metric measurements were discussed before and after the implementation of TMMI level 2 and Scrum. Accordingly, "Test Case Effectiveness" measurements were increased from 51% to 84%. Considering the "Defect Quality", it was increased from 69% to 89% after applying model. "Defect Density" was increased from 25% to 33%. In our study, "Test Case Effectiveness", "Defect Quality", and "Defect Density" measurement values increased from 65% to 80%, from 69% to 87%, and from 9% to 25%, respectively, which are represented in Table 4. It was observed that the two studies obtained similar results.

In the last part, the metric results of addressing TMMI level 3 and TMMI level 4 with the SAFE model will be compared. In view of the number of defects found on the basis of the project version, it is observed that the number of defects found before TMMI & agile process adaptation is higher, and the number of defects found upon the addition of new TMMI levels decreased. Even if it seems to be a negative picture in general, it is not accurate to evaluate this metric alone. With this metric, it is necessary to take the valid number of defects metric into account. It is understood in the light of these two pieces of data that 30% of the defects opened before this adaptation were either canceled or not accepted as defects. At the same time, in addition to these two metrics, the changes made on the code side on the basis of the project version and the number of added/changed files must not be overlooked. When the said three metrics are evaluated together, the number of valid defects in version 1 remains low compared to other levels, in view of the project version time. Serious differences are present between the first and other versions in the test effort and cost metric. According to this, the work done in the unit of 10 man/day, before TMMI and agile process adaptation, can be carried out via 5 man/day by reducing it by half. An important issue to pay attention to, here, is the difference between version 1 and other versions of the total number of test scenarios. The current number of 787 test scenarios was revised during the transformation process and the number of unnecessary tests was reduced by using risk-based testing approaches. Among the test scenarios, 77% were excluded from the scope through the transformation. This has shown us that if the tests are conducted without quality measurement and without being linked to a certain model, it inflicts unnecessary test effort and loss of resources. Defects related to testing scenarios; that is, test effectiveness, also indicate a positive increase. While there were many defects found in version 1 outside of the current test scenarios, this ratio, which was 65% in version 2, version 3, and version 4, increased to 80%. This shows us the importance of the methods used when reducing test scenarios with agile process transformation and the risk-based testing approach. Defect quality also indicates a significant increase in parallel with the TMMI versions. The correct opening of defects shows us that both the use of a test-oriented model and in-team communication with agile processes increase. When we take a look at risk-based metrics, both risk identification effectiveness and risk reduction effectiveness are shown as 0% because of the fact that this method was not used before TMMI and agile process transformation. With TMMI level 2 and agile process transformation, risks are identified in refinement meetings and integrated into the entire process. As can be seen in Table 4, a significant decline was observed in both risk identification effectiveness and risk reduction effectiveness during TMMI level 3 integration. It appeared that the reason for it was that risks were not addressed in the refinement meetings due to time pressure. With the retrospective meetings held under the agile process, this issue was discussed in both the run and the release retrospective meetings, and all the teams were informed about it; it was observed that this situation disappeared with the new version (TMMI level 4), and the effectiveness of risk reduction increased above 50%. In parallel with other metrics, improvements are observed in the number of postponed defects and

the number of defects coming from the customer compared to the period before the agile process–TMMI adaptation. Accordingly, the majority of the defects are rectified within that version and are not postponed to the next project version; in parallel to it, the defects ratios coming from the customer that could not be found in the testing stage drop by half.

This study and metrics demonstrate that agile processes and TMMI can work in harmony and have positive impacts on quality processes. With the use of TMMI, it was observed that the quality of test processes, found defects, and test scenarios have increased, team communication has enhanced thanks to agile processes, and project delivery dates have been shortened owing to agile process meetings. In addition to them, it was determined that the test effort and cost are significantly reduced by using risk-based testing approaches.

## 6. Conclusions

The study aimed to integrate agile process practices and the SAFE model with TMMI level 2, TMMI level 3, and TMMI level 4, as well as to reveal how much they can overlap. Thanks to this, by showing the strengths and weaknesses of agile processes and TMMI, complementary applications and risk-based testing approaches, to be used in order for those two approaches to work in harmony with each other, were mentioned.

When the results of the study were scrutinized, three different TMMI levels, which were attempted to be adapted, could be addressed with a rate of 90% and above with agile processes and risk-based testing approaches. In addition, with the metric measurements performed after each project version, the quality and test processes are clearly shown on the basis of the project versions. The effort exerted and the number of man/days prior to using TMMI and agile processes were halved. Positive increases were also observed in defect quality and defect ratios found in relation to the test scenarios. Thanks to this, by not postponing the defects in the version, faster action is taken, and a significant decline is observed in the number of defects returned from customers.

The main limitation of this study is the scarce number of studies in the literature to validate the results of our study. Another limitation may be metrics used in the analysis, evaluation, and report parts of the project life cycle that can be included in the project (test design efficiency, test review efficiency, etc.). On the other hand, the metric measurement frequency can be increased. In our study, measurements were made between project versions. This situation leads us to believe we cannot apply any statistical tests to validate the improvement between TMMI levels. As stated by the TMMI Foundation, this model can be adapted to both large and small organizations [8]. This study has only been validated on a large-scale organization. The same model can be applied in a small company, and the results can be compared.

For the future work, two more studies are planned to be conducted. They are planned to include TMMI level 5—Optimization level—the first of which was not addressed within the scope of this study, and to make the measurement of it with existing metrics in the new version of the project. In the first stage, it is planned to address the sub-titles of defect prevention, quality control, and test process optimization with agile processes and, then, to demonstrate the quality increase with the test metrics to be used under those conditions and at the end of the project. In the second study, it is planned to come up with a new three-level model after the integration of all TMMI levels with agile processes. With this model, which is entirely integrated with agile processes, it is considered to eliminate the TMMI level 4—Measured Value level—and distribute those sub-areas at all levels. In addition, it is planned to carry the non-functional tests, which are conducted below level 3, to below level 2, thanks to the risk-based testing approach.

**Author Contributions:** Conceptualization, A.U.; methodology, A.U.; software, A.U.; validation, A.U.; formal analysis, A.U.; investigation, A.U.; resources, A.U.; data curation, A.U.; writing—original draft preparation, A.U.; writing—review and editing, A.U.; visualization, A.U.; supervision, M.Ö.C.; project administration, O.K. All authors have read and agreed to the published version of the manuscript.

**Funding:** This research received no external funding.

**Conflicts of Interest:** The authors declare no conflict of interest.

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
