# Peer review of "Adaptation of the Four Levels of Test Maturity Model Integration with Agile and Risk-Based Test Techniques"

_electronics, doi:10.3390/electronics11131985_

Round 1
Reviewer 1 Report
I recommend a major revision based on the below points. Please, add a point to point response to each comment in your revision:
- I am not convinced about the novelty of the manuscript.
Don't use abbreviations in the title.
- The abstract is not technical and needs to clearly highlight the research gap.
- The abstract also missed statistical information about the results.
- The structure of the paper is vague. The paper needs to be restructured. Introduction section is very shorter and weak.
- Don't add heading over heading. Add a few lines related to the detail of a particular section before starting a sub-section.
- Proofread your paper from a native English speaker. There are many typos and grammar mistakes.
- At the end of the Introduction section, add the contributions clearly. See this paper for reference and citation ' A Novel Framework for Prognostic Factors Identification of Malignant Mesothelioma through Association Rule Mining.
- Related work/background/literature review should have a threat to a validity section. At the start of the background section, add a threat to a validity section. In that section, state the search strings and databases that have been explored to find the related work. See the below papers for references and citations 'Performance comparison and current challenges of using machine learning techniques in cybersecurity' and 'A Survey on Machine Learning Techniques for Cyber Security in the Last Decade'.
- The literature needs to be sub-divided into multiple sub-sections.
- Add the below papers to your literature: 'Taxonomy of automated software testing tools', Web Based Programming Languages that Support SELENIUM Testing, A comparative study of numerical analysis packages
- Add the statistical test (t-test/p-test/ANOVA, whichever is applicable) to compare your method with others.
- Comparison with the state-of-the-art is missed. You need to compare your method with the ground truth.
- You need to add a separate section for limitations of your approach and Future directions.
- It needs to add the reasons why these metrics are used for comparison.
-
Overall, the paper has many inconsistencies, and the contributions are not clear. The results are not compared with the ground truth properly. Limitations are not provided in their current approach. Future directions are not clearly stated.
I am looking forward to seeing your revised version.
All the best.
Reviewer 2 Report
Authors propose an adaption of four levels of TMMI based on agile and risk test techniques, and evaluate the influence of various factors for each level of TMMI through an agile process. Various testing metrics are used on the agile process for establishing the numerical analysis model.
The proposed approach looks sound and interesting. The paper is well organized and well presented. The proposed results and validations are encouraging. However, it is necessary to include a comparison with the recent existing testing methods without incremental adaptation. Furthermore, the authors must justify the adoption of the agile process in their proposed approach and give more details about the proposed analysis method.
In conclusion, the paper can be improved provided that the authors answer the above-mentioned questions in the new version
Reviewer 3 Report
The study aimed to integrate agile process practices and the SAFE model with TMMI 541 Level 2, TMMI Level 3, and TMMI Level 4, and to reveal how much they can overlap. The topic is interesting. Some comments are given as follows:
1. The background should be enriched in the introduction.
2. The related works should be summarized.
3. The description of Fig. 1 should be with more details.
Round 2
Reviewer 1 Report
The authors have addressed my comments. Congraturaltions.
Reviewer 2 Report
I recommend this paper to be accepted since authors have satisfactorily responded to all my questions and remarks.